# The Slow Progression of Diabetic Retinopathy Is Associated with Transient Protection of Retinal Vessels from Death

**DOI:** 10.3390/ijms241310869

**Published:** 2023-06-29

**Authors:** Yanliang Li, Basma Baccouche, Norma Del-Risco, Jason Park, Amy Song, J. Jason McAnany, Andrius Kazlauskas

**Affiliations:** 1Department of Ophthalmology and Visual Sciences, University of Illinois at Chicago, Chicago, IL 60612, USA; liyl@uic.edu (Y.L.); basma@uic.edu (B.B.); delrisc2@uic.edu (N.D.-R.); korsakoff@gmail.com (J.P.); amysong2@uic.edu (A.S.); jmcana1@uic.edu (J.J.M.); 2Department of Physiology and Biophysics, University of Illinois at Chicago, Chicago, IL 60612, USA

**Keywords:** diabetic retinopathy, protection from diabetic retinopathy, retinal capillaries, oxidative stress

## Abstract

The purpose of this study was to investigate the reason that diabetic retinopathy (DR) is delayed from the onset of diabetes (DM) in diabetic mice. To this end, we tested the hypothesis that the deleterious effects of DM are initially tolerated because endogenous antioxidative defense is elevated and thereby confers resistance to oxidative stress-induced death. We found that this was indeed the case in both type 1 DM (T1D) and type 2 DM (T2D) mouse models. The retinal expression of antioxidant defense genes was increased soon after the onset of DM. In addition, ischemia/oxidative stress caused less death in the retinal vasculature of DM versus non-DM mice. Further investigation with T1D mice revealed that protection was transient; it waned as the duration of DM was prolonged. Finally, a loss of protection was associated with the manifestation of both neural and vascular abnormalities that are diagnostic of DR in mice. These observations demonstrate that DM can transiently activate protection from oxidative stress, which is a plausible explanation for the delay in the development of DR from the onset of DM.

## 1. Introduction

Diabetic retinopathy (DR) is one of the microvascular complications of diabetes (DM). In 2020, the worldwide prevalence of DR in patients with DM was 22.27%, affecting about 103.12 million people [1]; this number is projected to increase to 160.50 million by 2045.

The diagnosis of DR is based on the morphological and functional status of the retinal vasculature. DR is classified as mild non-proliferative (NPDR) (microaneurysms only), moderate NPDR (any of the following: microaneurysms, retinal dot and blot hemorrhages, hard exudates or cotton wool spots;), severe NPDR (any of the following: intraretinal hemorrhage (≥20 in each of 4 quadrants), definite venous beading (in 2 quadrants) or intraretinal microvascular abnormalities) and proliferative diabetic retinopathy (PDR) (one or more of the following: neovascularization, vitreous or preretinal hemorrhages) with or without macular edema (retinal thickening or hard exudates) [2].

Patients who develop vision-threatening diabetic retinopathy (macular edema and proliferative diabetic retinopathy) are treated with intravitreal anti-VEGFs, laser photocoagulation, intravitreal corticosteroids or vitreoretinal surgery [3,4]. While all anti-VEGFs neutralize VEGF-A, aflibercept targets additional members of the VEGF family including VEGF-B and placental growth factor (PlGF) [5]. PlGF can promote inflammation, its level is elevated in the vitreous of patients with DR [6,7] and anti-PLGF antibodies prevent diabetic retinopathy in experimental animals [8]. Taken together, these results suggest that the beneficial effect of anti-VEGFs such as aflibercept include the suppression of inflammation that is driven by PLGF [9].

DR develops slowly and progressively after the onset of DM. The WESDR [10] showed that the prevalence of PDR was 0% at 3 years and increased to 25% at 15 years. Another study found that it took an average of 21 years for PDR to develop [11]. While understanding why the progression of DR is gradual has enormous therapeutic potential (e.g., indefinitely delaying vision-threatening DR), the underlying mechanism has not been intensively investigated.

In a small subset of patients with type 1 DM (T1D), DR is delayed for 50 or more years. The exceptional resistance to DR among the Joslin 50-Year Medalist Study [12] participants was not associated with strict glycemic control; the average HbA1c was 7.3 ± 1.0%. Rather, it appears that some other form of protection prevents DR in these individuals. The existence of Medalists demonstrates that delaying DR in the continued presence of DM is feasible.

Increased oxidative stress within the retina is one of the drivers of DR pathogenesis. In both patients [13] and experimental animals [14], prolonged hyperglycemia promotes the generation of reactive oxygen species (ROS), which increase oxidative stress. Furthermore, preventing oxidative stress protects experimental animals from DR. Overexpressing manganese superoxide dismutatse (MnSOD), an enzyme that scavenges superoxide suppresses both elevated oxidative stress within the mitochondria and the development of DR [15,16]. Similarly, the endothelial cell-specific knockout of NADPH oxidase 4 (Nox4) decreases mitochondrial ROS generation and protects the retina from acellular capillary formation in diabetic mice [17]. Finally, antioxidant-based therapies prevent DR in animals [18,19,20,21,22,23,24]. Even though there are many changes associated with DR other than increased oxidative stress [13,25], these studies demonstrate that controlling oxidative stress suffices to prevent diabetic animals from succumbing to DR.

The endogenous redox system includes a plethora of suppressors of oxidative stress. For instance, nuclear factor erythroid 2-related factor 2 (NRF2) is a redox-regulated transcription factor that governs the expression of many genes that mitigate oxidative stress [26,27]. Diabetic Nrf2 knockout mice [28] exhibit a reduced level of retinal glutathione and an early onset of blood–retina barrier dysfunction. The novel NRF2 activator dh404 attenuates oxidative stress and protects the vision-threatening breakdown of the blood–retinal barrier in diabetic rats [29]. These results indicate that an endogenous antioxidant system exists and that activating it can prevent diabetic retinopathy.

A hyperglycemia-driven increase in the level of oxidative stress within the mitochondria is believed to kill cells within retinal vessels and thereby play an essential role in the pathogenesis of DR in both patients and animal models (Figure 1A) [2]. While mechanistic studies with vascular cell types (endothelial cells and pericytes) indicate that the hyperglycemia-induced dysfunction of mitochondria is a rapid, self-amplifying and irreversible process [30], it takes years or even decades of DM for sight-threatening DR to develop in most patients with DM [10]. Taken together, these observations suggest the existence of a system that protects the retinal vasculature from oxidative stress-induced death, and DR develops only after the demise of this protective system.

The purpose of this study was to investigate the reason that DR is delayed in diabetic mice. To this end, we tested the hypothesis that the deleterious effects of DM are initially tolerated because endogenous antioxidative defense is elevated and thereby confers resistance to oxidative stress-induced death. We found that this was indeed the case in both T1D and T2D mice. Further investigation with T1D mice revealed that protection was transient; it waned as the duration of DM was prolonged. Finally, the loss of protection was associated with the manifestation of both neural and vascular abnormalities that are diagnostic of DR in mice. These observations demonstrate that DM can transiently activate protection from oxidative stress, which is a plausible explanation for the delay in the development of DR from the onset of DM.

## 2. Results

### 2.1. DM Transiently Increased Antioxidant Genes Expression within the Retina

We developed experimental systems to test if protection exists in the streptozotocin-induced, non-immune T1D mouse model of DR, which, like patients, does not develop DR coincident with the onset of DM. In this model, vascular and/or neural abnormalities become detectable within weeks, and their severity increases as the duration of DM extends to months [14,31,32]. If such a defense system exists, then its effect should inversely correlate with the severity of DR, i.e., protection will be greatest when retinal abnormalities are the slightest.

In light of the importance of oxidative stress in DR pathogenesis and the existence of endogenous antioxidant defense, we considered if protection involved an increased expression of antioxidant genes. qRT-PCR was performed on the total retina to assess the expression of genes that encode transcription factors (*Nrf2*) that govern antioxidant programs [33], NAD(P)H Quinone Dehydrogenase 1 (*Nqo1*), which encodes a cytoplasmic 2-electron reductase [34], mitochondrially localized enzymes that suppress oxidative stress (*Sod2*) [35] and the first rate-limiting enzyme responsible for the synthesis of the antioxidant glutathione (*Gclc*) [36]. We observed that the retinal expression of this panel of genes was markedly elevated after 5 days of DM (Figure 1B). After 20 weeks of DM, the expression of this panel of antioxidant genes declined or returned to baseline (Figure 1C). Thus, the duration of DM determines the direction of expression: short and long durations result in up- and downregulation, respectively.

### 2.2. An Approach to Detecting Ischemia/Oxidative Stress-Induced Death within Isolated Retinal Vessels

The following is the rationale for the design of the protection assay. Because a short duration of DM increased the expression of antioxidant genes (Figure 1B), we postulated that this change would confer resistance to oxidative stress-induced death. We focused on the retinal vasculature because death of cells within retinal vessels is a hallmark of DR pathogenesis. Figure 2A illustrates the key steps of this assay; Figure 2B displays representative images. The results of dose response experiments (Appendix A) indicated that 5 mM TBH induced a readily detectable degree of cell death in the vasculature isolated from the retina of healthy, non-DM mice.

### 2.3. The Vasculature of Diabetic Mice was Protected from Oxidative Stress/Ischemia-Induced Death

Having set up the protection assay, we used it to determine if the vasculature from mice that had experienced a duration of DM that causes only modest damage (5–8 days) was protected as compared with non-DM controls. There was a statistically significant decrease in the extent of death in the vasculature from DM versus non-DM mice in seven of eight sets of arbitrarily selected eyes (Appendix A). Similarly, the average fold change in death of all of the DM mice was lower as compared with that of the non-DM control mice (Figure 3). These observations demonstrate that the vasculature of mice that experienced DM for durations insufficient for readily detectable retinal abnormalities was protected from oxidative stress/ischemia-induced death.

To assess if DM-induced protection was unique to male mice, which were used in Figure 3, a similar series of experiments were conducted with female mice. We observed that a duration of DM that was insufficient to induce readily detectable DR [37] also triggered protection in female mice (Appendix A). There was a statistically significant decrease in the extent of death in five of six sets of arbitrarily selected eyes from non-DM and DM mice. Similarly, the average fold change in the death of all of the DM mice was lower as compared with that of the non-DM control mice (Appendix A). We conclude that DM increased the resistance of the retinal vasculature to oxidative stress/ischemia-induced death in both male and female mice.

### 2.4. DM Endowed the Retinal Vasculature with Protection from Cytokine-Induced Death

Having established that protection increased resistance to oxidative stress/ischemia-induced death, we considered if it extended to other types of DM/DR-related insults. For instance, inflammatory cytokines (IL-6, TNF-α and IL-1β) are elevated in the vitreous of patients with DR [38] and can kill vascular cell types [39]. The first step in this undertaking was to test if a short duration of DM increased the endogenous expression of cytokines. qRT-PCR analysis, which was focused on DR-associated cytokines and inflammation-related genes, demonstrated that expression was increased in the retina of DM mice (Figure 4A). Thus, a brief duration of DM, which did not cause blatant DR, nonetheless increased the expression of both cytokines and inflammation-related genes. However, there was no corresponding decline in the number of cells/unit area (cellularity) within the vasculature. Figure 4B indicates that either the magnitude of the increase in the level of cytokines was insufficient to kill vascular cells and/or protection included resistance to cytokine-induced death.

To further investigate if protection extended to cytokines, we established an assay to induce and detect cytokine-driven death of the retinal vasculature. To this end, we modified the nature and mode of delivery of the death-inducing insult that was used in the assay described in Figure 2. Instead of ex vivo administration of TBH/ischemia, a cocktail of cytokines (TNF-α, IL-1β and IFN-γ) was intravitreally injected into one eye; the other eye was injected with the cytokine vehicle (phosphate buffered saline (PBS)). Pilot experiments were performed to establish the minimum dose and duration of the cytokine cocktail that resulted in a readily detectable extent of cell death in the retinal vasculature of healthy non-DM mice (Figure 5).

To test if DM triggered protection from exogenous cytokine-induced death, non-DM and DM (5 days) mice were subjected to the assay shown in Figure 5. We observed that that a short duration of DM protected from exogenous cytokine-induced death; the cytokine cocktail caused less death in DM as compared with non-DM mice (Figure 6). Together, these data reveal that protection extended to resistance to cytokine-induced death.

Figure 2B and Figure 5D show that the appearance of TUNEL-positive nuclei depended on the insult that was used to induce death. The oxidative stress/ischemia insult resulted in cell apoptosis in a tram-track pattern, while the cytokine insult induced cell apoptosis in a discrete and sharply delineated pattern. Both patterns can be seen in retina vessels from DR patients [40], suggesting that both types of agents drive death in humans.

### 2.5. Prolonging the Duration of DM Was Associated with a Loss of Protection and the Appearance of Vulnerability

If protection prevents DR, then it should be lost as the duration of DM is extended to weeks and months, whereupon retinal abnormalities associated with DR manifest and intensify. Indeed, protection was no longer detected in mice that had experienced 20 weeks of DM (Figure 7 and Appendix A). In contrast, protection was present after 8 days, which, like 5 days of DM, is too short a duration for the development of many of the abnormalities associated with DR [14,31,32]. We conclude that the protection was transient and was lost as the duration of DM was extended.

The loss of protection (Figure 7), along with a decline in the number of neural and vascular cells, which has been previously reported [31], suggests that vulnerability to DM-related insults increases as the duration of DM is prolonged. The first step to test this concept was to measure the cellularity (number of cells/unit area) of the vasculature. While acellular capillaries arise from the complete absence of cells within a capillary segment, cellularity detects even a partial decline in cell number. We found that a short duration of DM had no effect (Figure 4B), whereas cellularity declined after 20 weeks of DM (Figure 7). These observations, made with un-insulted eyes, indicate that prolonging the duration of DM increased vulnerability to death driven by endogenous factors.

We used the protection assay (Figure 2) as a second approach to consider the effect of DM duration on vulnerability. As shown in Figure 7 and Appendix A, vulnerability to oxidative stress/ischemia was only observed in animals that experienced prolonged DM. Intermediate durations of DM showed a duration-dependent response; after 15 days of DM, protection and vulnerability were 20% and 0%, respectively; after 4 weeks of DM, protection and vulnerability were 25% and 25%, respectively (Appendix A). The fact that vulnerability and protection did not add up to 100% suggests that a loss of protection does not immediately result in a detectable increase in vulnerability.

Taken together, these results indicate that the duration of DM influenced the effect it had on the vasculature. While the finding that various parameters of retinal dysfunction intensify as the duration of DM is prolonged has been extensively documented by numerous investigators [14,31,32], we extend this body of knowledge with the novel observation that a short duration of DM, which does not cause overt DR, protected the retinal vasculature from DM-associated insults.

### 2.6. Loss of Protection and the Appearance of Vulnerability Was Associated with the Manifestation of DR

We also noted that the loss of protection and the appearance of vulnerability was associated with the development of DR. Animals that experienced a duration of DM sufficient for the manifestation of DR (16–20 weeks) displayed dysfunction of the ERG b-wave (Figure 8A), a decline in retinal thickness (Figure 8B and Appendix A), the appearance of acellular capillaries (Figure 8C) and an increased expression of permeability-related genes (Figure 8D). Taken together, these results indicate that the emergence of retinopathy was associated with a loss of protection and increased vulnerability.

### 2.7. A Brief Duration of DM Induced Protection in Type 2 Diabetic Mice

To determine if protection was unique to the T1D model, we repeated key experiments with the db/db type 2 diabetes (T2D) mouse model. In this series of experiments, db/db mice were DM for at least 6 days but not longer than 25 days (see Section 4), whereas the blood glucose of the db/+ mice was always below 250 mg/dL (Table 1). This duration of DM is insufficient for the development of most DR outcomes in this model [32]. The mice were sacrificed, one eye was subjected to the protection assay, whereas the expression of antioxidant genes was determined in the retina of the other eye. As shown in Figure 9, both protection and increased expression of key members of the antioxidant defense system were greater in db/db as compared with db/+ mice. We conclude that a duration of DM that does not cause overt DR increased the expression of the antioxidant defense system within the retina and protected the retinal vasculature from oxidative stress/ischemia-induced death in both T1D and T2D mouse models.

## 3. Discussion

The overall goal of this project was to investigate the reason for the delay between the onset of DM and the manifestation of DR. We observed that DM increased the resistance of retinal vessels to oxidative stress- and cytokine-induced death. Protection was transient and replaced by an increased vulnerability to death as the duration of DM was prolonged. Furthermore, the loss of protection and the appearance of vulnerability were temporally aligned with the manifestation of DR. These findings provide evidence for an endogenous system that protects the retinal vasculature from the deleterious effects of DM and thereby delays the manifestation of DR from the onset of DM.

In this study, we developed assays to detect protection from the oxidative stress-induced death of retinal vessels. Oxidative stress is an established driver of key pathogenic steps that are quintessential to DR such as the death of retinal capillaries. The use of these assays enabled the discovery of a previously unappreciated endogenous system that delays DM-related damage of the retina.

Our findings expand the current appreciation of how DM regulates the expression of antioxidant defense genes. The expression of these genes is suppressed in the retinas of mice that experience durations of DM sufficient to cause DR [16,41]. We report that DM can also increase expression and that the duration of DM determines the direction of the change: a short and long duration of DM results in up- and downregulation, respectively. This new discovery aligns with previous reports indicating that the overexpression of *Sod2* alone suppresses mitochondrial oxidative stress, mitochondrial dysfunction and DR [16,41]. Together, these findings suggest that protection exists and that it involves defense from DM-driven oxidative stress.

The role of oxidative stress in the pathogenesis of DR is widely recognized, and the mechanism by which it occurs has been aggressively investigated. Chronically elevated blood glucose, a hallmark of DM, results in a multitude of metabolic abnormalities, including the activation of the polyol and hexosamine pathways, increased protein kinase C (PKC) activity and an accumulation of advanced glycation end products (AGEs) [25,42]. In addition, the NADPH oxidase (NOX) complex is activated and thereby increases cytosolic ROS production in both neural and vascular cell types within the retina [23]. The sustained elevation of ROS within the cytoplasm promotes mitochondrial dysfunction, which includes the disruption of the electron transport chain (ETC), increased production of the superoxide and damage to the mitochondrial DNA (mtDNA). These changes result in a vicious cycle of increased free radical production, which amplifies oxidative stress in a self-perpetuating and irreversible manner [30] (Figure 1A). The leakage of cytochrome c from dysfunctional mitochondria triggers apoptosis of cells within the retinal vasculature, which is a diagnostic feature of DR [23]. An endogenous protective system would be expected to counter events that drive DR pathogenesis, such as increased oxidative stress.

The success of antioxidant therapy in preventing DR supports the concept that endogenous protection from DR involves the suppression of oxidative stress. Supplementing the diet of Goto-Kakizaki rats (a type 2 diabetic model) with a combination of vitamins C and E for 36 weeks prevents hallmarks of DR, including acellular capillaries and pericyte ghosts [24]. Similarly, supplementing the diet of T1D rats with lipoic acid for 44 weeks inhibits mitochondrial dysfunction and capillary cell apoptosis [22]. Furthermore, the administration of the polyphenol antioxidants found in green tea suppresses the DM-driven decline of the endogenous antioxidant defense system (including SOD, GSH and CAT) and protects from DR [20]. Common features of antioxidant treatments that prevent DR in animals include starting the treatment close to the onset of diabetes, prolonged treatment and measuring robust, death-related outcomes within the vasculature.

While the efficacy of antioxidant supplements in protecting patients from DR remains controversial, some show promising results, especially those using a combination of antioxidants. For example, a 5-year follow-up study by García-Medina et al. [43] showed that the oral administration of Vitalux Forte^®^, containing vitamins C and E, lutein, b-carotene and trace elements, significantly reduced plasma lipid peroxidation end products and slowed the progression of DR in patients with T2D with NPDR. Another study using Nutrof Omega^®^, containing vitamins C, D, B and E, omega-3, lutein, glutathione and trace elements, for 18 months showed a significant reduction in total oxidant stress, plasma lipid peroxidation by-products and decreased DR progression and onset [44]. Similarly, the oral administration of Nutrof Omega^®^ for 38 months significantly reduced pro-oxidants, increased antioxidants and slowed DR progression in patients with T2D with NPDR [45]. The profiles of patients who benefit from these types of antioxidant supplements are T2D without DR or mild-to-moderate NPDR without DME or with DME but without the thickening of the retina. These findings suggest that antioxidants have the potential to be beneficial when administered during the early stages of DR, when anatomical damage is not excessive, but they may not be able to reverse damage once it has occurred [46,47].

While protection from oxidative stress-induced death is likely to involve the increased expression of antioxidant defense genes, previous publications suggest that suppressing oxidative stress may also prevent cytokine-induced death. The NRF2 pathway, known to play a critical role in cellular protection against oxidative stress [26,27,28], also acts as a negative regulator of inflammation and mitigates inflammation-associated pathogenesis in various diseases [48]. Furthermore, the activation of Nrf2 for 10 weeks in STZ-induced diabetic rats reduces the levels of TNF-α and IL-6 in the retina [29]. Additional investigation is required to assess if protection, which attenuates the cytokine-induced death of the retinal vessels in the T1D mice, involves more than an antioxidant-driven mechanism.

Studies with participants of the Joslin 50-Year Medalist Study revealed that the mechanism of durable protection from DR in patients was associated with retinol binding protein 3 (RBP3), a retinol transport protein expressed mainly by photoreceptors. A higher level of RBP3 in the vitreous was associated with less severe diabetic retinopathy and a reduced risk of developing proliferative diabetic retinopathy (PDR) in participants of the Medalist Study [49]. RPB3 associates with glucose transporter 1 (GLUT1) to decrease glucose uptake and thereby attenuate the hyperglycemia-driven expression of inflammatory cytokines [49]. Since RPB3 is also an antioxidant [50,51,52], its elevated expression may suppress oxidative stress and thereby enforce the endogenous protection that is described in this report.

If the endogenous system of protection requires persistent hyperglycemia, then it may explain the phenomenon of “early worsening” in DR [53]. In the Diabetes Control and Complications Trial (DCCT), the cohort with tight glucose control experienced a worsening of DR in the first year [54], although they ultimately had better outcomes. If the reduction in blood glucose resulting from tight glucose control shuts off protection, then the progression of retinopathy would accelerate. Additional experimentation is necessary to address this intriguing possibility.

In conclusion, we discovered the existence of an endogenous system that protects cells within the retinal vasculature from DM-associated death. Protection was transient and replaced by increased vulnerability to death, which temporally coincided with manifestation of DR. These findings suggest that protection delays the onset of DR, which commences only upon the deterioration of protection. Unveiling the mechanism of protection will enable novel therapeutic approaches to delaying DR indefinitely.

## 4. Materials and Methods

### 4.1. Animals

Seven-week-old male and female C57/BL6/J (Jax #000664) mice and six-week-old db/db and db/+ male mice (BKS.Cg-Dock7m+/+Leprdb/J; Jax #000642) were purchased from Jackson Lab, Bar Harbor, ME, USA. The animals were housed in group cages in a pathogen-free environment on a 12-light/dark cycle and were provided free access to food and water. The animals were euthanized by CO2 asphyxiation, and the eyes were enucleated and processed immediately. All animal studies were approved by the Office of Animal Care and Institutional Biosafety at the University of Illinois at Chicago.

### 4.2. T1D Mouse Model

Diabetes was induced in C57/BL6/J mice with five daily consecutive intraperitoneal injections of streptozotocin (Sigma Cat#: S0130, Sigma-Aldrich, Burlington, MA, USA) in 47 mM sodium citrate (pH 4.5). The dose of streptozotocin used in male and female mice was 60 and 75 mg/kg/day, respectively. Age-matched non-diabetic animals received injections of an equal volume of citrate buffer. Fasting blood glucose was measured two days after the last dose of STZ. Mice with fasting blood glucose levels greater than 250 mg/dL for three consecutive days were considered diabetic, with the first day considered diabetic day 1 (Appendix A). Male mice were used at 5, 8 and 15 days and 4, 16 and 20 weeks after the onset of DM. Female mice were used 8 days after the DM onset. The short (5 and 8 days) and long (16 and 20 weeks) durations of DM were chosen because they are not, or are, respectively, of a sufficient duration for the development of DR [14,31,32]. The body weight and fasting blood glucose were monitored weekly and daily, respectively.

### 4.3. T2D Mouse Model

The db/db and db/+ mice arrived when they were 42 days old. Five days later we started measuring their unfasted blood glucose every other day. Those db/db mice whose blood glucose was greater than 250 mg/dL for four consecutive readings were sacrificed and the eyes were analyzed as described for the T1D mice. The age-matched db/+ mice, whose blood glucose was always below 250 mg/dL, were processed in parallel. At harvest, all mice were 53 days old. Thus the db/db mice experienced DM for at least 6 days (assuming that the onset of DM coincided with the first time that the blood glucose was measured), but no longer that 25 days (assuming the DM developed on day 28)

### 4.4. Protection Assay

The protection assay was used to detect insult-induced death within isolated retinal vessels. This assay is a modification of a previously published procedure [55]. Briefly, freshly enucleated eyeballs were put into DMEM with 1% BSA with or without Tert-butyl hydroperoxide (TBH, sigma, #75-91-2) for 1 h at 37 °C and then fixed with 10% formalin overnight. The dose of oxidative stress (TBH) was chosen to induce a readily detectable level of cell death within the retinal vessels of non-DM mice. The retina was dissected, incubated overnight in water and then digested with trypsin (3% trypsin 1:250) (Amresco Solon, OH, USA, 0458-50G) for 4.25 h at 37 °C. After the neuro-retinal tissue was gently brushed away, the isolated vascular bed was transferred to a glass microscope slide and allowed to air dry (Figure 2A). Next, the vasculature was permeabilized and stained with IB4 (Thermo, Waltham, MA, USA, I32450) overnight at 4 °C and then stained with TUNEL (Roche, Santa Clara, CA, USA, #11684795910) and mounted using a DAPI-containing mounting medium (Invitrogen, Waltham, MA, USA #P36935). Confocal images of the resulting vasculature were obtained using a Zeiss LSM 710 fluorescence microscope (Zeiss Microscopy, Inc., White Plains, NY, USA); six to eight randomly selected fields surrounded the optic nerve. The number of apoptotic bodies (TUNEL/DAPI double-positive species) was counted using Image J Fiji (https://imagej.nih.gov/ij/, NIH, Bethesda, MD, USA). In this series of experiments, the magnification was either 20× or 40×, and the actual size of the captured photos was 425 μm × 425 μm or 360 μm × 360 μm, respectively. Sets (a pair of eyes; one from a randomly selected non-DM and a second from a DM mouse) were processed in parallel. The fold change is the average number of apoptotic bodies in one DM eye divided by the average number in one non-DM eye within one set.

The retinal vasculature from un-insulted eyes was stained with periodic acid–schiff hematoxylin (PASH). Images were photographed with a bright-field microscope. A total of 8–10 40× pictures were taken in the peripheral region around the optic nerve. The number of nuclei and acellular capillaries per visual field (425 μm × 425 μm) was counted using Image J and manually, respectively.

### 4.5. Intravitreal Injection (IVT)

Mice were anesthetized with an intraperitoneal injection of 100 mg/kg ketamine and 5 mg/kg Xylazine. All drugs used with animals were obtained from the University of Illinois at Chicago Pharmacy. Intravitreal injections were performed with a 33-gauge needle customized by Hamilton (Reno, NV, USA); the injection site was 2–3 mm from the limbus; the volume was 1 μL/eye. For each mouse, the left eye was injected with phosphate buffered saline (PBS), whereas the right eye was injected with a cytokines cocktail in PBS. At the 1× concentration, the cocktail consisted of a 1:1:1 ratio of 100 μg/mL recombinant human TNF-α (Peprotec, Cranbury, NJ, USA; # 300-01A), 100 μg/mL recombinant human IL-1β (Peprotec; # 200-01B) and 1500 U/μL recombinant human IFN-γ (Peprotec; #300-02). At the desired time point after receiving the injection (typically, 24 h), the mice were euthanized, their eyes were enucleated and the retinal vasculature was isolated and stained as described above. The number of apoptotic bodies throughout the entire retinal vasculature was counted manually.

### 4.6. qRT-PCR

The animals (*n* = 8–10/group/time point) were sacrificed, the eyes were enucleated and the retinas were dissected. The total RNA from the retinal tissue was extracted and purified according to the manufacturer’s instructions using the RNeasy mini kit (Qiagen, Germantown, MD, USA). The RNA concentration and purity were determined via spectrophotometry (Implen NanoPhotometer™ NP80, Westlake Village, CA, USA). In total, 1 μg of total RNA was reverse-transcribed with a cDNA-synthesis kit (HighCapacity cDNA Reverse Transcription Kit) (Thermo Fisher, Waltham, MA, USA). QRT-PCR experiments were performed using the QuantStudio™ 7 Flex system (Applied Biosystems^®^, Waltham, MA, USA) with Fast SYBR Green Master Mix (Applied Biosystems^®^). For normalization and relative quantification, we used ct values of the housekeeping gene *Actb.*

### 4.7. Optical Coherence Tomography (OCT)

OCT was performed with the Micron IV system (Phoenix Research Labs, Pleasanton, CA, USA). The mice were anesthetized with an intraperitoneal injection of a mixture of 100 mg/kg ketamine and 5 mg/kg Xylazine. The pupils were dilated with 1% tropicamide eye drops. All drugs used with animals were obtained from the University of Illinois at Chicago Pharmacy. For each eye, four OCT sections were imaged (superior, inferior, temporal, nasal) 350 μm from the optic nerve head. The corneal surface was protected using polyethylene glycol and propylene glycol eye gel during and after the imaging process. The thickness of the following layers was measured and averaged in both eyes with the InSight2D software (Phoenix Research Labs, Pleasanton, CA, USA): ganglion cell complex (GCC), including the nerve fiber layer–ganglion layer, inner plexiform layer), inner nuclear layer (INL), outer plexiform layer (OPL), outer nuclear layer (ONL), inner segment/outer segment (IS/OS) and total retina. For each OCT image, three points were selected to represent the thickness of the retina. These points were chosen to not include blood vessels, which overestimate the total retinal thickness. The data obtained from both eyes of the same animal were averaged.

### 4.8. ERG Methods and Analysis

Mice were dark-adapted for 2 h prior to the ERG recordings and anesthesia (100 mg/kg ketamine; 5 mg/kg xylazine), which was administered by intraperitoneal injection under dim red illumination. The pupils of the mice were fully dilated with tropicamide (1%) drops. All drugs used with animals were obtained from the University of Illinois at Chicago Pharmacy. Stimuli were generated and delivered using a Celeris rodent electrophysiology system (Diagnosys LLC, Lowell, MA, USA). This instrument makes use of a combined stimulator/electrode probe that was placed in contact with the cornea of each eye, aligned with the center of the pupil. A conductive gel (Systane) was placed on the tip of the electrode. Stimuli were delivered to one eye at a time, with the fellow, unstimulated eye serving as the reference. The flash stimuli consisted of brief (≤4 ms), short-wavelength (453 nm), full-field luminance that ranged from −6.0 to 1.0 log cd-s-m-2. A minimum of three flash responses for each stimulus luminance were obtained and averaged for analysis. From these mean responses, the a- and b-wave amplitudes and implicit times were calculated according to convention.

### 4.9. Statistical Analysis

The two-tailed student’s *t*-test was performed to assess the statistical difference between two groups using Prism 9.0 (GraphPad Software, Irvine, CA, USA). The results are shown with the mean ± SEM. *p* < 0.05 was considered a significant threshold. One-way ANOVA was used to calculate the statistical difference between three or more groups.

## Figures and Tables

**Figure 1 ijms-24-10869-f001:**
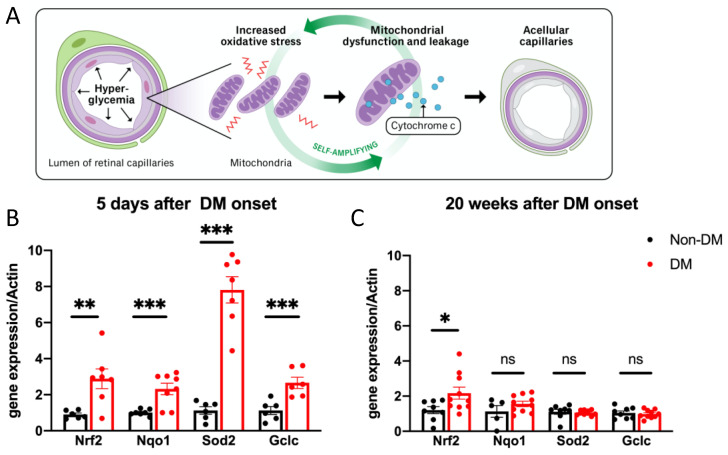
DM induced the transient expression of antioxidant genes within the retina. (**A**) Diagram of the current dogma for how hyperglycemia increases oxidative stress and thereby causes the death of cells within retinal capillaries. We postulate that DR is delayed from the onset of DM because of an endogenous system that suppresses HG-driven oxidative stress and thereby protects from the deleterious effects of DM. (**B**) Total retinal tissue was isolated from mice that had been DM for 5 days or age-matched non-DM mice and was subjected to qRT-PCR analysis. Bar graphs represent a change in the message levels of the indicated genes (red bar; the data from each eye from 6–10 mice) and non-DM retinas (black bar; the data from each eye from 6–10 mice), normalized to β-actin. The data are expressed as the fold increase over non-DM mice and represent the mean ± SEM. ** *p* < 0.01; *** *p* < 0.005. (**C**) Same as (**B**) except that the mice experienced a duration of DM that suffices for the development of DR (20 weeks) [31]. * *p* < 0.05, “ns” the differences is not statistically different.

**Figure 2 ijms-24-10869-f002:**
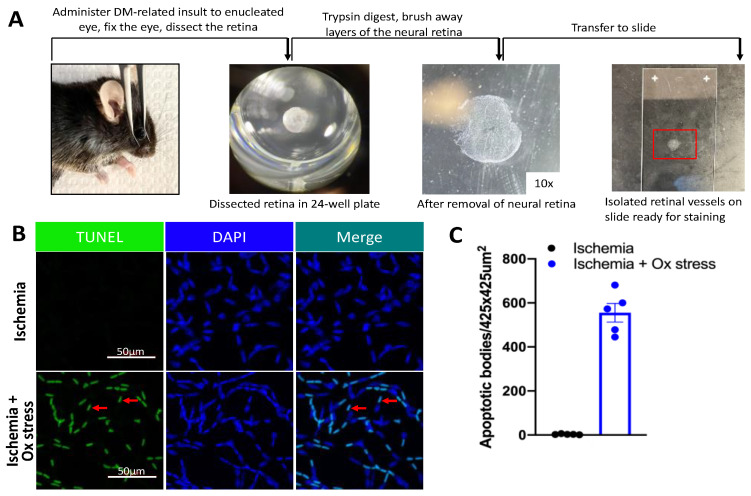
An approach to detect ischemia/oxidative stress-induced death within isolated retinal vessels. (**A**) Key steps of the assay for detecting death within retinal vessels. (**B**) Representative images of ischemia ± ox stress-induced apoptotic bodies in isolated retinal vessels. The red arrows point to representative TUNEL/DAPI double-positive species. (**C**) Quantification of results in (**B**). Five randomly selected regions (425 μm × 425 μm) in the peripheral zone around the optic nerve within a retina were selected and photographed. The number of apoptotic bodies in each region was counted with Image J and is represented by a dot in the bar graph. An apoptotic body is defined as a TUNEL/DAPI double-positive species. The data in panel (**C**) are from a single mouse. The same results were observed on at least five independent occasions with five additional mice.

**Figure 3 ijms-24-10869-f003:**
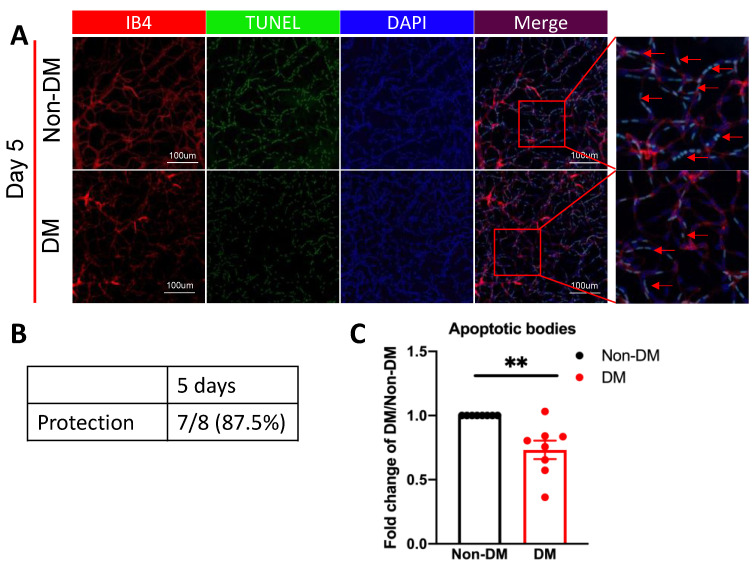
The vasculature of diabetic mice was protected from ox stress-induced death. (**A**) Representative images of ox stress-induced apoptotic bodies in the isolated retinal vasculature 5 days after the onset of DM. The red arrows point to representative IB4/TUNEL/DAPI-positive species. (**B**) Sets (a pair of eyes from randomly selected non-DM and DM mice) were processed in parallel. Protection: a statistically significant smaller number of apoptotic bodies/unit area in the vasculature of DM versus non-DM eyes within a given set. (**C**) Within each set of eyes, the average number of apoptotic bodies within six to eight visual fields in one DM eye was divided by that of a non-DM eye to determine the fold change. (*n* = eight sets.) The fold change of the DM group and age-matched non-DM group is presented. Data shown are the mean ± SEM. ** *p* < 0.01 with student’s *t*-test.

**Figure 4 ijms-24-10869-f004:**
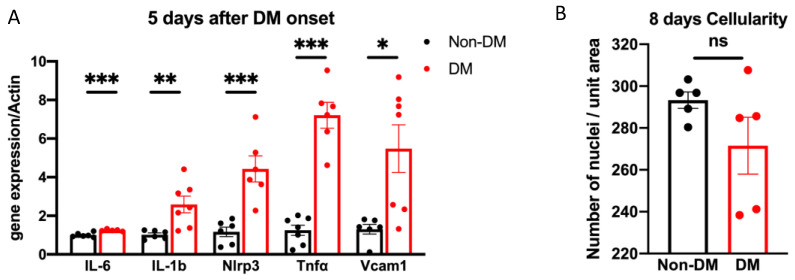
The expression of inflammation-related genes was elevated by DM. (**A**) Total retinal tissue was isolated from mice that had experienced 5 days of DM and then subjected to qRT-PCR analysis. Bar graphs represent changes in the message levels of inflammatory-related gene expression for DM (red bar, *n* = 10) and non-DM retinas (black bar, *n* = 8), normalized to β-actin. * *p* < 0.05; ** *p* < 0.01; *** *p* < 0.005. (**B**) Retinal vessels from mice that had experienced 8 days of DM along with age-matched non-DM control mice were isolated from un-insulted eyes and stained with periodic acid–schiff hematoxylin (PASH). For each eye, 8–10 arbitrarily selected regions (360 μm × 360 μm) within the peri-optic nerve zone were selected and photographed. The number of nuclei was counted with Image J; the average for each eye is shown as a dot in the bar graph. Five non-DM and five DM mice were analyzed. “ns” the differences is not statistically different.

**Figure 5 ijms-24-10869-f005:**
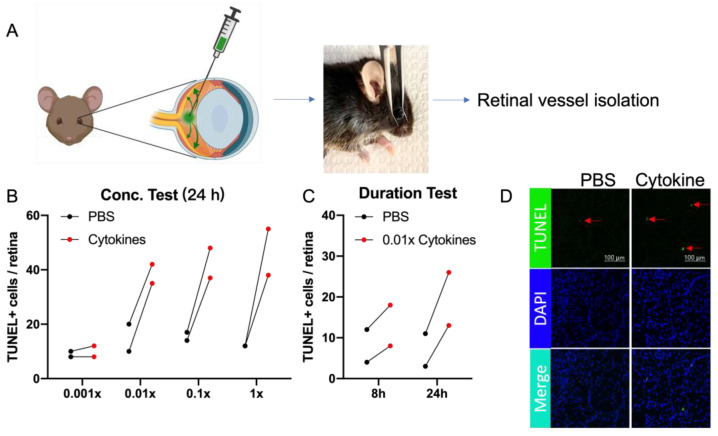
Detection of cytokine-induced death within retinal vessels. (**A**) Key steps of the assay for observing death of retinal vessels resulting from the intravitreal injection of a cytokine cocktail. (**B**) Dose-response to cytokines-induced death. 1× cytokines cocktail consisted of a 1:1:1 ratio of 100 ug/mL TNF-α, 100 ug/mL IL-1β and 1500 U/uL IFN-γ. Each mouse was injected intravitreally with PBS in the left eye and cytokines in the right eye. The number of apoptotic bodies (TUNEL/DAPI double-positive species) in the entire retinal vasculature was counted manually. (**C**) Duration-response to cytokines-induced death. Each mouse was injected intravitreally with PBS in the left eye and 0.01× cytokines in the right eye for 8 h or 24 h, respectively. The extent of cell death was quantified as described in panel (**A**); pilot experiments shown in panels (**B**,**C**) were conducted on two mice. (**D**) Representative images of a subset of the total retinal vasculature from a single mouse are shown. The red arrows point to representative TUNEL-positive species.

**Figure 6 ijms-24-10869-f006:**
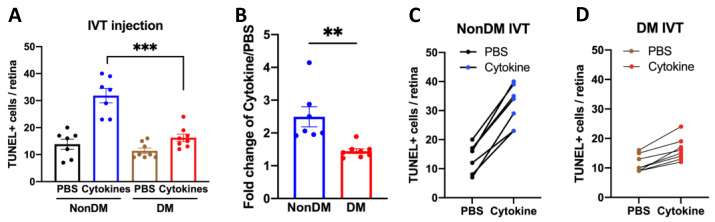
The vasculature of diabetic mice was protected from cytokine-induced death. (**A**) A total of 5 days after the DM onset, both groups (DM, *n* = 8; Non-DM, *n* = 7) were injected intravitreally with PBS in the left eye and 0.01× cytokines in the right eye. After 24 h, the eyes were harvested and analyzed as described in Figure 5. The extent of death was not different in PBS-injected eyes. Each point is the average number of TUNEL+ cells in a single eye. *** *p* < 0.005. (**B**) The average cytokine cocktail-induced fold change in the entire group of mice for each experimental condition. The bar graph displays the mean ± SEM. The student’s *t*-test was used to assess if the differences between groups were statistically significant; ** *p* < 0.01. (**C**) The raw data for each non-DM mouse. (**D**) The raw data for each DM mouse.

**Figure 7 ijms-24-10869-f007:**
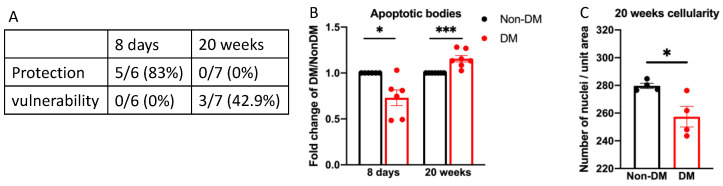
Prolonging the duration of DM was associated with a loss of protection and the appearance of vulnerability. (**A**) Mice that experienced the indicated duration of DM, along with age-mated non-DM control mice, were subjected to the assay described in Figure 2. Six and seven sets of eyes were used for the 8 day and 20 week cohorts, respecitively, where a set consists of two eyes, one from a DM mouse and one from a non-DM mouse. Protection: a statistically significantly smaller number of apoptotic bodies/unit area in the vasculature of DM versus non-DM within a given set. Vulnerability: a statistically significantly larger number of apoptotic bodies/unit area in the vasculature of DM versus non-DM within a given set. (**B**) The fold change in the number of apoptotic bodies (TUNEL/DAPI double-positive species) for the mice described in panel (**A**). The 20-week data were corrected for the cellularity by dividing the number of apoptotic bodies by the average decline in cellularity shown in panel (**C**). The 8-day data were not corrected because the cellularity of the vasculature was unchanged after 8 days of DM (Figure 4B). * *p* < 0.05; *** *p* < 0.005. (**C**) Retinal vessels from the un-insulted eyes were obtained from mice that had experienced 20 weeks of DM along with non-DM controls (*n* = 4) and processed as described in the legend of Figure 4.

**Figure 8 ijms-24-10869-f008:**
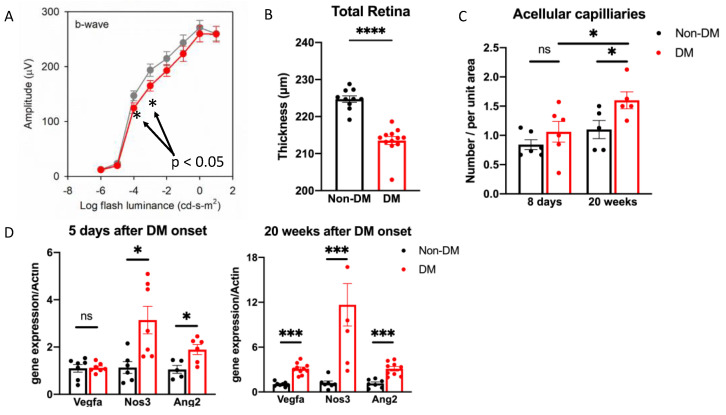
Retinopathy was associated with a loss of protection and increased vulnerability. (**A**) The b-wave of ERG from mice that experienced DM for 16 weeks (red circle, *n* = 12) compared with that of age-matched non-DM mice (grey circle, *n* = 10). (**B**) The quantification of total retina thicknesses in DM (*n* = 12) and non-DM (*n* = 10) mice. In these experiments, mice were DM for 16 weeks. **** *p* < 0.0001. (**C**) The number of acellular capillaries was counted manually in retinal vessels from animals that had experienced DM for the indicated duration; “non-DM” indicates age-matched control animals. * *p* < 0.05. (**D**) qRT-PCR was performed as described in the legend of Figure 1 to assess the expression of vascular leakage-related genes in the whole retina from mice that experienced the indicated duration of DM (5 days; *n* = 6–8) and age-matched non-DM mice (*n* = 8). Note that the *y* axis scale bar differs in the two graphs. * *p* < 0.05; *** *p* < 0.005, “ns” the differences is not statistically different.

**Figure 9 ijms-24-10869-f009:**
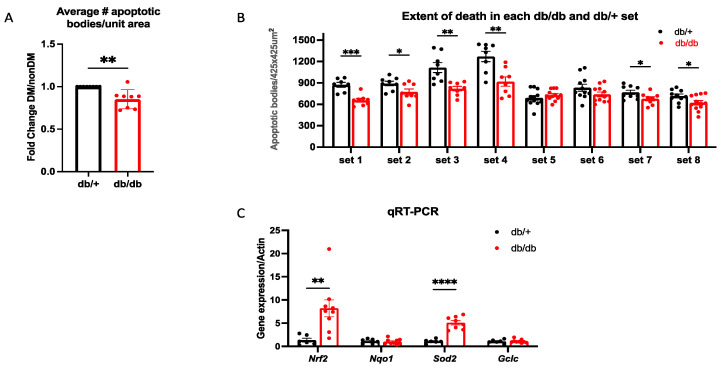
Key discoveries in the T1D model were also observed in T2D mice. db/db mice that experienced DM for at least 6 days and age-matched db/+ mice (that were non-DM) (*n* = 8) were subjected to the protection assay described in Figure 2. (**A**) The average fold change between all eight sets of db/+ and db/db eyes. ** *p* < 0.01. (**B**) The data for each of the eight sets. * *p* < 0.05; ** *p* < 0.01; *** *p* < 0.001. (**C**) The expression of the indicated genes was assessed as described in the legend of Figure 1; *n*= 8–10. The bar graphs display the mean ± SEM; *t*-test for statistical significance. ** *p* < 0.01; **** *p* < 0.0001.

**Table 1 ijms-24-10869-t001:** Blood glucose (mg/dL); mean ± SD.

Age (Days)	47	49	51	53
db/db; *n* = 10	500.0 ± 30.4	472.1 ± 90.3	482.0 ± 50.3	>250 mg/dL; sacrificed
db/+; *n* = 8	186.9 ± 25.4	173.3 ± 27.3	162.8 ± 24.9	<250 mg/dL; sacrificed

## Data Availability

Data are available upon reasonable request.

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
