# Peer review of "The Slow Progression of Diabetic Retinopathy Is Associated with Transient Protection of Retinal Vessels from Death"

_ijms, 2023, doi:10.3390/ijms241310869_

Round 1

Reviewer 1 Report

Authors have identified a cell-protective pathway that is activated in the very early stages of DR that looks protrective compared to non-diabetic controls.

There are a few typos and minor grammatical and punctuation errors that can be corrected by additional proofreading by the authors. In the later stages of DR, this, however, turns pathological causing cell stress and death. While the data is robust, the significance of the study is weak. Nevertheless, there is a novelty in the study. It was hard to follow the content and the concept at times. hence, providing a schematic of the working hypothesis would be ideal for this manuscript.

Author Response

Reviewer #1

Comments and Suggestions for Authors

Authors have identified a cell-protective pathway that is activated in the very early stages of DR that looks protrective compared to non-diabetic controls.

Comments on the Quality of English Language

There are a few typos and minor grammatical and punctuation errors that can be corrected by additional proofreading by the authors.

As requested, we corrected the typos, grammatical and punctuation errors. 

In the later stages of DR, this, however, turns pathological causing cell stress and death. While the data is robust, the significance of the study is weak. Nevertheless, there is a novelty in the study. It was hard to follow the content and the concept at times. hence, providing a schematic of the working hypothesis would be ideal for this manuscript.

We edited and/or re-wrote multiple sections of the manuscript to improve its clarity.  These changes include articulation of the project’s working hypothesis (see text related to Figure 1A).

Please note that the revisions include addition of a new data showing that key discoveries made with the T1D model were also observed with a T2D model (Figure 9 and Table 1). Thus protection manifests in mouse models of both types of DM. The Abstract, Materials and Methods, Introduction, Results and Discussion sections have been modified accordingly.  

Reviewer 2 Report

Please find the comments and suggestions in the attached word file.

As written in my "comments and suggestions to the authors", generally, the English is of good quality. However, the wording and phrase structures are often complex and difficult to understand. Additionally, there are some misspelling and grammatical errors. Thus, a careful revision of the manuscript is recommended.

Author Response

Reviewer #2

The authors presented an article that analyzed the association of the activation of the endogenous antioxidant defense system to oxidative stress as well as inflammation (in terms of increased inflammatory cytokines) in diabetic mice related to the duration of diabetes to reveal potential causes for the “delayed” onset of diabetic retinopathy considering the duration of diabetes.

The manuscript describes a large study with multiple complementary analyses and good sample sizes. However, the hypothesis is not convincing, the study design is not appropriate to corroborate it, and the conclusions are not convincing as well.

Nevertheless, the data can add valuable new knowledge to the pathogenesis of diabetic retinopathy, if it is differently interpreted. It has to be noted that the use of male and female mice is appreciated as well as the transparent presentation of data as bar graphs and dot plot.

In the following it is tried to explain in detail the hard decision, followed by the usual list of minor and major comments.

If understood correctly, the scientific question posed and made conclusions are that DR often manifests as a late complication in diabetes, though detrimental processes theoretically start to act immediately (e.g., hyperglycemia is inducing oxidative stress that should “immediately” induce cell damage). However, it is known that often, diabetic retinopathy develops years after onset of diabetes.

Thus, the study aimed to find a reason for this delayed onset of diabetic retinopathy.

The data demonstrated an activated antioxidant defense and decreased apoptosis after oxidative stress insult in eyes of diabetic mice. Additionally, data showed an increased expression of inflammation-related genes and decreased apoptosis after cytokine insult in eyes of diabetic mice. The authors conclude that diabetes activates an endogenous antioxidant and anti-inflammatory defense systems that protect cells from oxidative stress and inflammation in early diabetes; this protection gets lost in late DM. It is written that 1) “we discovered the existence of an endogenous system that protects cells within the retinal vasculature from DM-associated death” and 2) “These findings suggest that protection delays the onset of DR, which commences only upon the deterioration of protection”, and that “Unveiling the mechanism of protection will enable novel therapeutic approaches to delay DR indefinitely”.

The conclusions cannot be made:

  • The authors did not discovered a new The antioxidant and anti-inflammatory defense mechanisms are well known and it is not surprising that they are activated in case of oxidative stress and inflammation. Moreover, the experiments mainly analyzed cell damage in terms of rate of apoptosis and retina thickness but did not revealed an underlying mechanism.
  • It is questionable if “ [DM-related] protection delays DR onset”, since no DM mice without oxidative-/inflammation-insult were analyzed and thus, there is no information about oxidative stress/inflammation level before insult. It is assumed that a statement in the sense that “a (progressively) weakened endogenous defense can delay but finally not prevent DR onset” seems to better reflect the Additionally, such statements are difficult without knowledge about the underlying mechanism (see also 1).
  • “absolute” statements like “when the mechanism will be known it will be possible to delay DR indefinitely” are generally Additionally, such statements are quasi-impossible without knowledge about the underlying mechanism (see also 1).

The 3 main limitations leading to aforementioned conclusions are the following:

  • The analyses done are measuring both – oxidative stress level and the protection – with same methods. Thus, the information cannot be clearly separated and differentially analyzed. However, if the purpose is to analyze a protective mechanism against oxidative stress, it is key

1

to determine protection and oxidative stress separately. Otherwise, a lower rate of apoptotic cells, e.g., can mean a lower level of oxidative stress or it means a higher protection against (still high levels) of oxidative stress, what are obviously different conclusions.

  • The project compares only diabetic mice to non-diabetic mice after insult (oxidative stress or cytokines). However, to see, if diabetes is activating a protective mechanism other than the “normal” antioxidant defense, it is crucial to measure oxidative stress level in eyes from diabetic mice without It could be assumed that oxidative stress is already increased without additional insult and thus, the activation of the antioxidant defense would be expectable.
  • As already mentioned, most experiments were done in “insulted” eyes from diabetes non- diabetes eyes. However, the in vivo analyses (OCT and ERG) examined non-insulted animals and thus, this data cannot be “mixed” with the other data (even they are “only” shown to confirm diabetic retinopathy onset at 16 weeks of diabetes duration). This is a completely different treatment group.

To summarize, the present study is not appropriate to answer the scientific question. However, the data can nevertheless be valuable, if presented differently. A proposition would be to analyze in more detail the different time points (that are currently shown only in the supplement). The manuscript could be restructured and rewritten in the following sense:

  • The authors analyzed the time course/phases of cellular degeneration/damage/vulnerability in diabetes that finally lead to diabetic

We are not able to complete this series of experiments in the time frame permitted for revisions.  However, we cite the work of other groups that have documented the time course/phase of cellular degeneration/damage using the same T1D model that was used in this manuscript.  

  • Analyses were done in eyes of diabetic mice compared to eyes of non-diabetic mice in which oxidative stress/inflammation were induced (or better increased?); additional analyses were done in living diabetic mice compared to non-DM mice – without additional

We re-wrote multiple sections of the manuscript to clarify the design and purpose of the experiments, and also to highlight the novelty of the discoveries.  For instance, Fig 1B shows that expression of multiple members of the endogenous anti-oxidant defense system increases after 5 days of DM in the retina of un-insulted eyes.  These analyses were done with un-insulted eyes.  Consequently,  the upregulation of the endogenous anti-oxidant system resulted from DM rather than an insult.  To the best of our knowledge, this has not been previously reported. 

            The discovery that DM increased expression of key members of the endogenous anti-oxidant defense system suggested that the retina would be resistant to oxidative stress.  We proceeded to test this hypothesis in a separate series of experiments shown in Figures 2 and 3.  We compared the extent of insult-induced death in the vasculature of non-DM and DM mice, and found that the vasculature from DM mice was resistant to oxidative stress/ischemia-induced death.  To the best of our knowledge, DM-driven acquisition of protection has not been previously reported. 

            Together these finding provide a plausible explanation for why the retinal vasculature resists oxidative stress/ischemia-induced death.  Namely, DM upregulates the endogenous anti-oxidative defense system and thereby provides protection from oxidative stress that results from chronic hyperglycemia. 

  • Data show a functional endogenous defense that protects the diabetic retina (i.e., hyperglycemic) from oxidative stress-/inflammation-driven damage in the early phase of DM. This defense seems to weaken continuously with prolonged duration of diabetes until a breakpoint is reached and cell damage The first phase lasts probably around 1 week; the second phase lasts from 2-4 weeks (if interpreting sFigure 5 correctly, the defense seemed lowered compared to week one, but significant damage is neither seen); in phase 3 that started around week 16, cell damages became significant and diabetic retinopathy manifested.

(this is of course only a very roughly worded proposition to transport the idea).

We edited multiple sections of the manuscript to better articulate the following three concepts

  1. A) Protection was transient.
  2. B) As the duration of DM increased, protection was replaced with increased vulnerability to insult-induced death.
  3. C) Loss of protection and increased vulnerability coincided with manifestation of DR.

Then, the manuscript should be newly submitted.

Minor comments

  1. Please unify the format (e.g., spaces between title and text of a figure caption) and modify it to improve readability (e.g., to start every new line of figure captions with an indentation disrupts reading flow).

As requested, the figure legend format has been modified to improve its readability.

  1. Please revise the wording of the manuscript that sometimes is too “strong” (e.g., line 110: DM regulates antioxidant genes) or too “complex” (e.g. lines 120-122: “The following is the rationale for the design of the protection assay. Because we anticipated that protection involved increased expression of anti-oxidant genes, we chose oxidative stress as the insult with which to induce death”. This could be shortened to: “a protection assay has been designed to evaluate the expression of antioxidant genes in response to an oxidative stress insult to induce cell death”.)

We edited the manuscript to institute this request.  Please note that we were unable to incorporate the suggested wording because it did not accurately describe the experiments that were done.  More specifically, the effect of DM (rather than oxidative insult) on expression of anti-oxidant genes was evaluated.  Oxidative insult was administered to detect protection. 

  1. Please correct misspelling and grammatical

We corrected misspellings and grammatical errors throughout the manuscript.

  1. Please exchange the term “apoptotic bodies” by “apoptotic cells”, since apoptotic bodies are a distinct type of apoptotic extracellular vesicles and this is not what has been

We respectfully request permission to retain the “apoptotic bodies” term because we are uncertain if they arise from a single cell.  The oxidative stress/ischemic insult resulted in fragmentation of the nuclei (Figure 2B). Since a single nucleus results in multiple fragments, the term “apoptotic cells” would incorrectly indicate the number of cells present in a capillary.  We modified the text related to the “apoptotic bodies” term in multiple place to clearly define this term and thereby articulates its intended meaning in the present context.   

  1. Line 108: Please correct “expect” to “except”.

This correction has been made.

  1. The name “protection assay” seems It has been chosen to emphasize that a protective mechanism was planned to be revealed. However, the assay induced damage and (mainly) analyzed cell death; this should be reflected in the name.

We respectfully request permission to retain the “protection assay” name  because it  reinforces the central theme of the manuscript.   The overall goal of this project was to advance our understanding of why manifestation of DR is delayed from the onset of DM in animal models.  Our working hypothesis is that there is an endogenous system that protects from the deleterious effect of DM, and that its deterioration is a prerequisite for development of DR.  The assay’s name reinforces the key concepts of the working hypothesis, and the nature of the outcomes that were investigated.  We are optimistic that the edited version of the manuscript will clarify the overall goal and  design of the experimental approaches, in which case the current name of the assay will no longer be confusing. 

  1. Often, the unit of the apoptosis analysis is “apoptotic bodies/425x425um2”. I suggest simplifying this by detailing the area in the methods and to write in the figures, g., “apoptotic cells/field”.

Please see our response to point #4. In addition, we modified the relevant text within the Results, Methods and Materials sections and/or figure legends.

  1. Please correctly write units and symbols (e.g., µm, ?, superscript ®, …).

This correction has been made throughout the manuscript. 

  1. Please add the magnification used in the figure

In the methods section (line 421 and 428) it is written “…6-8 20x pictures”. Is 20x the

magnification? Please explain the information.

As requested, we added the following information to the “Assay to detect ischemia/oxidative stress-induced death within isolated retinal vessels” section of the Materials and Methods

In this series of experiments the magnification was either 20x or 40x and the actual size of the captured photos was 425um x 425um or 360um x 360um, respectively.

  1. Figures should be understandable as a Please add all necessary information to figure captions.

The requested information has been added.

  1. Figure 4: Please explain the “unit area”.

The requested information has been added to the legend of Figure 4B.

  1. Line 249: Please explain the abbreviation “PASH”.

We modified the text to explain this term.

  1. Line 376: Please explain the abbreviation “DCCT”.

We modified the text to explain this term.

  1. Lines 433-434: Please add the ordering information for Ketamine and

We added the requested information.

  1. Lines 443-444: It is written “retinal vasculature were counted …by two researchers…”. Please explain how the data of the researchers were Do the results show the average of both analyses?

We modified the text to eliminate this ambiguity.  While the results of a single grader are presented, both graders observed protection from cytokine-induced death.

  1. Line 472: Please add the ordering information for

We added the requested information.

  1. Line 487: Please delete “Patents”.

Deleted.

  1. Lines 489-490: Please complete the information for the

The information for the Supplementary Materials has been completed. 

  1. Lines 495: Please delete “Please add”.

Deleted.

  1. Line: 499: Please delete “Name of Institute”.

Deleted.

Major comments

General

  1. Considering aforementioned comments, the whole study should be re-analyzed, re-structured, and

As requested, we revised the manuscript to embrace the aforementioned comments.

  1. Rate of apoptosis should always be presented in the same way: either the “fold change of

DM/non-DM” or the “number of apoptotic cells/area” to enable comparison.

The data emanating from a single set (one eye from a non-DM and one eye from a non-DM mouse) is presented as the # of apoptotic bodies/unit area. When the results of the all of the sets are presented, we use Fold change of DM/nonDM.  These two approaches are helpful to distinguish the two ways of presenting the data: as individuals sets and as the average of entire sets.  We revised the manuscript to clarify this point (see legend of Figure 2 and the legends of sFigures 1 and 2.

  1. The use of male and female mice is appreciated; however, why females were only analyzed on day 8? If possible, all time points should be repeated in female mice – or female mice should be not mentioned in this manuscript (and the fact mentioned as study limitation). Moreover, why the females were analyzed at 8 days and the males at 5 days? The difference is not explained and prevents comparability of

Generally, it is not clear how the time points were chosen; this should be explained in the methods section.

The requested information has been added. We define both 5 and 8 days of DM as a duration that is insufficient for most investigators to detect retinal abnormalities.  Such publications are cited in the revised manuscript. 

Title

  1. No correlations were calculated and thus, the title is misleading and should be

The title has been modified.

Introduction

  1. Overall, the introduction is well written except the purpose of the study that is not well explained (and should anyway modified).

Thank you for the positive assessment of the Introduction.  We modified it to clarify the overall purpose of the study. 

Results

  1. Often, the results include conclusions that belong to the discussion (e.g., lines 110-116). On the other hand, many results are described too vague/short or not described at all (e.g., line 124: “Figure 2B shows representative images”). Please rewrite the results

The Results section has been rewritten as requested.

  1. sFigure 1: It is written that a t-test has been done; however, 4 groups were Thus, the analysis should be repeated by performing an ANOVA (or the respective non-parametric analysis). It is also written that 4 eyes/group were analyzed; is this equal to 4 animals? Please, specify.

We modified sFigure 1 so that only two experimental conditions are presented.  The t-test was used to assess statistical significance between the two groups. 

Furthermore, we expanded the legend to include the requested information regarding the number of mice used in these experiments.

  1. Figure 5: it is written that the images show the “entire retina”. However, the images do not show tissue borders; is it really the entire retina? Please,

Thank you for  bringing this issue to our attention.   We  revised the legend of Figure 5 to explain this point. 

  1. Figure 2: Is it correct that the analysis in ‘C’ was made only in n=1 animal? Then, no statement about apoptosis in these conditions can be The figure should be deleted or shown in the supplement.

Figure 2C quantifies the relevant data shown in the Figure 2B image. In this series of pilot experiments the same results were observed on at least 5 independent occasions with 5 additional mice.  This additional information has been added to the legend of Figure 2.

  1. Figure 6: it is written that a t-test was However, for Figure 6 C and D a paired t-test would be appropriate (if statistics were made for this data).

We revised Figure 6 by including brackets to indicate the pairwise comparisons that we did using the t-test. 

  1. Lines 233-234: It is written that “The oxidative stress insult resulted in cell apoptosis in a tram- track pattern, while the cytokine insult induced cell apoptosis in a discrete and sharply delineated pattern”. The first is not visible in shown images and for the latter, no images are presented. Please add a figure to justify the

We re-wrote this portion of the manuscript to point out the figures that illustrate the distinct pattern of apoptosis (Figures 2B and 5D). 

  1. Figure 7 (and following figures that show similar results): the percentages of “protection” and “vulnerability” do not sum up to 100%. Should that not be the case? Please

They may not sum up to 100% if loss of protection does not immediately result in a detectable increase in vulnerability. We added this explanation to the text.

Discussion

  1. Lines 331-333: it is written that Kumar et found a declined antioxidant system in DM-rats at 48 h post-induction (in rats not-treated with green tea). Such an early decline would be the opposite to your results. Please discuss this controversy.

We re-wrote this portion of the Discussion to address this issue. Key components of the AOX defense system were suppressed in rats after 16 weeks of DM. Treating the DM rat with green tea for the entire duration of DM attenuated the decline and prevented hallmarks of DM.  These findings support our working hypothesis that the endogenous AOX system protects from DR. 

  1. Lines 337-353: The paragraph describes available antioxidant supplements to treatment/prevention diabetic retinopathy and describes the patient population that is probably benefitting most from such a Please link this data to the results of the study – treatment seems to be most beneficial in early phases, when the endogenous defense is still active, I guess.

The final sentence of this paragraph in the revised manuscript is pasted below

These findings suggest that antioxidants have the potential to be beneficial when ad-ministered during the early stages of DR, when anatomical damage is not excessive, but they may not be able to reverse damage once it has occurred 39, 40.

Methods

  1. Statistics: It is recommended to present data as mean±SD.

Only parametric tests were performed. Was a normality test done to confirm normal data distribution? If not, this should be caught up, or please justify why parametric were used.

We respectfully request permission to present data as mean ± SEM.  While we routinely use both to express variability, we used mean ± SEM in our initial experiments, and continued with this option throughout the study.  Importantly, normal (Gaussian) distribution was confirmed with the Shapiro-Wilk normality test using GraphPad (Prism 9.0.0).

  1. Line 406: Animals with a loss of body weight of >10% were insulin Were these mice still included in the analyses? The additional treatment let assume an exclusion. Please explain.

None of the mice lost more that 10% of their body weight in this study.  Consequently, we removed this statement from the Methods and Materials section. 

Please note that the revisions include addition of a new data showing that key discoveries made with the T1D model were also observed with a T2D model (Figure 9 and Table 1). Thus protection manifests in mouse models of both types of DM. The Abstract, Materials and Methods, Introduction, Results and Discussion sections have been modified accordingly.  

Reviewer 3 Report

This project aimed to consider the existence of endogenous systems that are likely responsible for the slow progression of DR. The authors focused on the protection of retinal capillaries from death induced by DM-associated insults such as oxidative stress and inflammatory cytokines.

Comments

Model development-

1.     Line 404-405.. typo. Please correct “also”.

2.     Can authors please provide references for the STZ doses they use, as recently it has been established that female mice need more STZ dosage compared to the male mice? Saadane et al. 2020.

3.     Figure 6- supplementary figure, is rather very important as it provides the evidence for the successful model development for all cohorts, unfortunately, the figure reference is not in the main text either in the methods section or in the result. Can the authors please mention sFigure6 in the main text?

4.     Can the authors please comment on why there are varied early time points like 5 and 8 days and why there is no consistent timeline of 5 days for an “early timepoint”?

5.     Can the authors please comment on why there are varied late time points like 16 weeks and 20 weeks and why there is no consistent timeline of 20 weeks for a “late timepoint”?

6.     Line 487- what does “6. Patents” refer to?

Introduction and Results-

1.     Figure 1-The authors have specified the “n”, but I would like to know whether the gene expression data was derived from the average of mechanical triplets or only biological replicates (n).

2.     In Figure 1, the Nqo1 gene and other antioxidant genes are shown in the graph, but Nqo1 is not described in the text.

3.     Can authors add references to the antioxidant genes description, lines 92-97

4.     I think the authors do not need to refer to #24 reference for lines 114-115..reference#9 itself emphasizes that overexpression of sod2 is protective.  

5.     For the development of protection assay, can authors show a line graph for the dose-dependent effect of TBH in sfigure1. 5mM concentration shows a significant increase in apoptotic cells, then why the authors don’t show the significance in the graph? 5mM is compared to 10 and 25mM, why is 5mM not compared to 2mM. It would be better if only 2mM and 5mM comparisons are shown as it emphasizes the dose-dependent effect. Can the authors please add information about TBH and why they chose TBH in methods or supplementary text?

6.     Figure 2 can authors please specify the “n” in the figure legend?

7.     In Figure 3, the resolution of the image is very low for DM retinas, and hard to visualize the difference in representation.

8.     How many mice are within the 6-8 sets for Figure 3?

9.     Figure 4- Represents the expression of inflammatory markers after 5 days of the onset of DM, but the figure legend mentions 20 weeks. I believe it is a typo. Please clarify the correct timeline.

10.  Consistent timelines are missing in throughout the article, can authors please comment on why 8 days to count cells in Figure 4? Also, males are counted at 5 but females (sFigure 3) are counted at 8 days.

11.  In Figure 5, the resolution of the TUNNEL image is low, and the representation of TUNNEL-positive cells is not clear. The earlier TUNNEL images are much better. The “n” is not specified in the figure legend for Figure 5 and the graphs do not have individual data sets. Was the dosage decided based on a single mouse per concentration?

12.  Suggestion- Since one eye is injected with PBS and the other eye is injected with Cytokine IVT, can the data be shown in a single graph with all four columns together for Figures 6A and 6B?

For example, black, and blue columns, space brown and red columns?

13.  As per Figure 7B, why are only 20 weeks of data corrected by a decline in cellularity and not 8 days when counting apoptotic bodies?

14.  As mentioned before for the DR pathology, why have the authors considered 16 weeks and not 20 weeks?

15.  Suggestion for Figure 8, can authors show 5 days versus 20 weeks differences in the ERG, total retinal thickness, and acellular capillaries? Figure 8D shows a similar comparison.

16.  Specifically for ERGs, literature has shown that changes in the a and b waves are seen in STZ mice after 5 months or more, so 20 weeks might be a better time point.

17.  Suggestion- authors have repeatedly mentioned differences in cellularity (endothelial cell and pericytes); counting acellular capillaries, although a hallmark measure of DR seems redundant here.

Author Response

Reviewer #3

This project aimed to consider the existence of endogenous systems that are likely responsible for

the slow progression of DR. The authors focused on the protection of retinal capillaries from death induced

by DM-associated insults such as oxidative stress and inflammatory cytokines. Comments

Model development-

  1. Line 404-405.. typo. Please correct “also”.

We were unable to find this typo.  However, we are optimistic that we corrected it during the thoroughly re-read and editing of the manuscript. 

  1. Can authors please provide references for the STZ doses they use, as recently it has been established that female mice need more STZ dosage compared to the male mice? Saadane et al. 2020.

This reference has been added and cited as requested.

  1. Figure 6- supplementary figure, is rather very important as it provides the evidence for the successful model development for all cohorts, unfortunately, the figure reference is not in the main text either in the methods section or in the result. Can the authors please mention sFigure6 in the main text?

Thank you for bringing this error to our attention.  We revised the text of the Results section to mention sFigure 6. 

  1. Can the authors please comment on why there are varied early time points like 5 and 8 days and why there is no consistent timeline of 5 days for an “early timepoint”?

We added an explanation for why both 5 and 8 days of DM were used.  Namely, both of these durations of DM precede most of the detectable outcomes associated with DR. Such publication are cited in the revised manuscript. 

  1. Can the authors please comment on why there are varied late time points like 16 weeks and 20 weeks and why there is no consistent timeline of 20 weeks for a “late timepoint”?

We added an explanation for why both 16 and 20 weeks of DM were used.  Namely, both of these durations of DM suffice for manifestation of most of the detectable outcomes associated with DR. Such publication are cited in the revised manuscript. 

Line 487- what does “6. Patents” refer to?

“6. Patents” has been deleted.

Introduction and Results-

  1. Figure 1-The authors have specified the “n”, but I would like to know whether the gene expression data was derived from the average of mechanical triplets or only biological replicates (n).

We modified the legend of this figure to clarify that a single eye from 6-10 mice was used in each of the two experimental groups. 

  1. In Figure 1, the Nqo1 gene and other antioxidant genes are shown in the graph, but Nqo1 is not described in the

We modified the text to include a description of the Nqo1 gene.

  1. Can authors add references to the antioxidant genes description, lines 92-97

The requested references have been added. 

  1. I think the authors do not need to refer to #24 reference for lines 114-115..reference#9 itself emphasizes that overexpression of sod2 is

We agree that the two papers make similar points. We respectfully request permission to cite both because it provides full credit to previous publications, and also because this concept is central to the overall theme of the manuscript. 

  1. For the development of protection assay, can authors show a line graph for the dose-dependent effect of TBH in sfigure1. 5mM concentration shows a significant increase in apoptotic cells,

then why the authors don’t show the significance in the graph? 5mM is compared to 10 and 25mM, why is 5mM not compared to 2mM. It would be better if only 2mM and 5mM comparisons are shown as it emphasizes the dose-dependent effect.

We revised sFigure 1 as requested; the new figure shows only the 2 and 5 mM doses of TBH, and that there was a statistically significant difference between 2 and 5 mM.

  1. Figure 2 can authors please specify the “n” in the figure legend?

Figure 2C quantifies the relevant data shown in the Figure 2B image. In this series of pilot experiments the same results were observed on at least 5 independent occasions with 5 additional mice.  This additional information has been added to the legend of Figure 2.

  1. In Figure 3, the resolution of the image is very low for DM retinas, and hard to visualize the difference in

We replaced the images in Fig 3A with one that more clearly demonstrates the key points.

  1. How many mice are within the 6-8 sets for Figure 3?

Each set consists of one eye from a non-DM mouse and a second eye from a DM mouse.  Thus 6-8 non-DM mice and 6-8 DM mice were used to generate the 6-8 sets described in Figure 3. The revisions of the manuscript have clarified this point. 

  1. Figure 4- Represents the expression of inflammatory markers after 5 days of the onset of DM, but the figure legend mentions 20 weeks. I believe it is a typo. Please clarify the correct

Thank you for catching this typo, which we corrected. 

  1. Consistent timelines are missing in throughout the article, can authors please comment on why 8 days to count cells in Figure 4? Also, males are counted at 5 but females (sFigure 3) are counted at 8

We added an explanation for why both 5 and 8 days of DM were used.  Namely, both of these durations of DM precede most of the detectable outcomes associated with DR. Such publication are cited in the revised manuscript. 

  1. In Figure 5, the resolution of the TUNNEL image is low, and the representation of TUNNEL positive cells is not The earlier TUNNEL images are much better. The “n” is not specified

in the figure legend for Figure 5 and the graphs do not have individual data sets. Was the dosage decided based on a single mouse per concentration?

As requested, we made the following changes to Fig 5 and its legend.

We replaced panel D with higher resolution images.

We revised the presentation of the data in panels (b) and (c) to indicate that two mice were used for each experimental condition in this series of pilot experiments.

  1. Suggestion- Since one eye is injected with PBS and the other eye is injected with Cytokine IVT, can the data be shown in a single graph with all four columns together for Figures 6A and 6B?

For example, black, and blue columns, space brown and red columns?

As requested, we combined panels A and B. 

  1. As per Figure 7B, why are only 20 weeks of data corrected by a decline in cellularity and not 8 days when counting apoptotic bodies?

The 20 wk data was corrected for cellularity because at this time point the number of cells/unit area declined in the retinal capillaries.  No such decline occurred at the 8 day time point (Figure 4B), and hence no correction was necessary.  We modified the text to clarify this point. 

  1. As mentioned before for the DR pathology, why have the authors considered 16 weeks and not 20 weeks?

We added an explanation for why both 16 and 20 weeks of DM were used.  Namely, both of these durations of DM suffice for manifestation of most of the detectable outcomes associated with DR. Such publication are cited in the revised manuscript. 

  1. Suggestion for Figure 8, can authors show 5 days versus 20 weeks differences in the ERG, total retinal thickness, and acellular capillaries? Figure 8D shows a similar

We are not able to complete the ERG and total retinal thickness experiments in the time frame permitted for revisions.  However, we cite the work of other groups that reported no change in the ERG after 1 week of DM in this same STZ mouse model.  Note also, that Figure 8C shows that there is no increase in acellular capillaries after 8 days of DM.  The text has been modified accordingly. 

  1. Specifically for ERGs, literature has shown that changes in the a and b waves are seen in STZ mice after 5 months or more, so 20 weeks might be a better time

We agree that an a wave abnormality  is likely to develop as the duration of DM is prolonged.  However, we are not able to complete such ERG analyses in the time frame permitted for revisions. 

  1. Suggestion- authors have repeatedly mentioned differences in cellularity (endothelial cell and pericytes); counting acellular capillaries, although a hallmark measure of DR seems redundant here.

Cellularity (# of cells/unit area of the capillary) was determined to accurately measure protection/vulnerability (# of TUNEL/DAPI double positive bodies/unit area).  We found that as the duration of DM increased, cellularity declined and hence the protection/vulnerability data was corrected accordingly.  Acellular capillaries (relatively short and narrow stretches vessels within the capillary bed that are completely devoid of cells) is a distinct outcome.  We modified the text to define each precisely and to point out that they are not the same parameter.  

Please note that the revisions include addition of a new data showing that key discoveries made with the T1D model were also observed with a T2D model (Figure 9 and Table 1). Thus protection manifests in mouse models of both types of DM. The Abstract, Materials and Methods, Introduction, Results and Discussion sections have been modified accordingly.  

Round 2

Reviewer 2 Report

The manuscript improved significantly and the effort done by the authors is highly appreciated! 

However, unfortunately, I still do not agree to the conclusions made. I am still thinking that the study design is not appropriate to confirm the proposed hypothesis and that consequently the results do not support the conclusions made (as written in the first review). 

Author Response

The manuscript improved significantly and the effort done by the authors is highly appreciated!

Thank you for this positive assessment of the revised manuscript.

However, unfortunately, I still do not agree to the conclusions made. I am still thinking that the study design is not appropriate to confirm the proposed hypothesis and that consequently the results do not support the conclusions made (as written in the first review).

We further revised the text of the manuscript to clarify the study design, results and conclusions. Such changes included a complete re-write of the abstract, as well as edits throughout the manuscript.